# Validation of a Multi-Residue Analysis Method for 287 Pesticides in Citrus Fruits Mandarin Orange and Grapefruit Using Liquid Chromatography–Tandem Mass Spectrometry

**DOI:** 10.3390/foods11213522

**Published:** 2022-11-04

**Authors:** Xiu Yuan, Chang Jo Kim, Raekeun Lee, Min Kim, Hee Jeong Shin, Leesun Kim, Won Tae Jeong, Yongho Shin, Kee Sung Kyung, Hyun Ho Noh

**Affiliations:** 1Residual Agrochemical Assessment Division, Department of Agro-Food Safety and Crop Protection, National Institute of Agricultural Sciences, Wanju 55365, Korea; 2Department of Applied Biology, College of Natural Resources and Life Science, Dong-A University, Busan 49315, Korea; 3Department of Environmental and Biological Chemistry, College of Agriculture, Life and Environment Science, Chungbuk National University, Cheongju 28644, Korea

**Keywords:** pesticide, multi-residue, QuEChERS, mandarin orange, grapefruit, liquid chromatography–tandem mass spectrometry (LC–MS/MS)

## Abstract

Since the introduction of the positive list system (PLS) for agricultural products in the Republic of Korea, the demand for a quick, easy multi-residue analysis method increased continuously. Herein, the quick, easy, cheap, effective, rugged, and safe (QuEChERS) technique combined with liquid chromatography–tandem mass spectrometry was employed to optimize a method for the multi-residue analysis of 287 pesticide residues in mandarin orange and grapefruit. Method validation was conducted in terms of selectivity, limit of quantitation (LOQ), linearity, accuracy, precision, and matrix effect. All the compounds at low spiking levels (1, 2.5, 5, or 10 mg/kg) could be quantified at LOQs lower than 0.01 mg/kg (PLS level). The linearity of the matrix-matched calibration curve for each compound is in the range 0.5–50 μg/L, and its coefficient of determination (*R*^2^) is >0.990. Satisfactory recovery values of 70–120% with a relative standard deviation of ≤20% are obtained for all compounds in the mandarin orange and grapefruit samples. A negligible matrix effect (−20–20%) is observed for more than 94.8% and 85.4% of the pesticides in mandarin orange and grapefruit, respectively. Therefore, this analytical method can contribute to pesticide residue analyses of citrus fruits for routine laboratory testing.

## 1. Introduction

Citrus fruits, oranges, grapefruits, and mandarin oranges, are among the most widely consumed fruits in the world [1,2]. Many species of fungi and insects infest citrus fruits, causing irreversible damage, leading to losses in crop yields [3]. Pesticides increase the production of agricultural products, such as citrus fruits. Pesticides can move through air, soil, and water over long distances, which risks their incorporation into crops even when not sprayed [4]. Consequently, pesticide residues are frequently detected in agricultural products, affecting food quality and human safety. In addition, they can cause several health risks and deleterious effects on consumers [3,5,6,7].

To address this problem, many regulatory agencies in various countries have established maximum residue limits (MRLs) in agricultural products used for consumption [8]. The MRLs of pesticide residues vary from country to country according to food consumption and agricultural and storage practices [9]. For international trade, food exporters must supply their products according to the standards of the importing country. Since January 2019, the PLS was introduced and implemented with a uniform limit (0.01 mg/kg) for agricultural products with no registered MRLs in the Republic of Korea [10]. Therefore, the demand for developing a multi-residue method to detect the unintentional contamination of various agricultural products by pesticides, including those not registered in the MRLs, is increasing. The analytical method must be able to determine many pesticide residues at the same time to ensure the quality of agricultural products. Currently, such a method is applied to monitor real samples [10,11].

In general, analytical methods are required for the qualitative and quantitative determination of unknown pesticide residues in samples. Gas chromatography (GC) and high-performance liquid chromatography (HPLC) are the commonly used methods; however, these universal techniques are unable to determine multiple residues simultaneously [12]. To compensate for this shortcoming, the monitoring of multiple residues in various matrices has been conducted using tandem mass spectrometry (MS/MS) [13]. MS/MS is widely used for the multi-residue analysis of food samples and environmental matrices, such as vegetables [14], fruits [15], mealworms [16], ginseng [17], and soil [18], as well as biological samples [19,20]. Moreover, MS/MS is widely used in pharmaceutical and medical sciences to analyze multiple essential components, including essential oils [21] and biomarkers [22]. 

Validation involves experimental designing to confirm that a method can produce results that are accurate within the scope of its intended use. To obtain reliable results during validation, both the recovery and matrix effect should be evaluated [13]. Direct analysis of some pesticide residues using LC–MS/MS is possible because multiple reaction monitoring (MRM) is a highly specific and sensitive technique that can selectively quantify compounds in complex samples. However, it usually requires a combination of partition and clean-up procedures to obtain the best recovery and remove the worst interferences [23]. 

Electrochemical techniques are currently popular for pesticide analysis. Nonetheless, LC–MS/MS is also advantageous because of its sensitivity and selectivity and the short analysis time [24,25]. However, LC–MS/MS suffers from drawbacks owing to the matrix effect [26]. Matrix effects due to the use of the electrospray ionization (ESI) technique must be evaluated during method development. These effects can considerably affect the sensitivity and reproducibility of trace analysis of the compounds [27]. The results of quantitative analysis using LC–MS/MS are reliable only when the matrix effects are assessed or compensated. The addition of an isotopically labeled internal standard for each compound is the best way to compensate for matrix effects. However, this is not feasible for simultaneous multi-residue analysis. Consequently, the matrix-matched calibration method has become the most common approach to compensate for matrix effects because it is relatively cheap and simple. However, it is difficult to find the same blank matrix to compensate for matrix effects [23,28]. 

Mandarin orange is one of five representative crops (besides rice, potato, pepper, and soybean) that are analyzed by government agencies using the multi-residue standardized method [10]. However, for other citrus fruits such as grapefruit, no standardized method has been developed. Chung et al. report that among all fruits in the Republic of Korea, the highest pesticide content (83.9%) is detected in citrus fruits. Among citrus fruits, grapefruit has the highest pesticide content (above 90%) [29]. A few studies were conducted on compensating matrix effects in multi-residue determination in citrus fruits. Hence, in this study, mandarin orange and grapefruit were selected as representative citrus fruits for developing a multi-residue method.

Recently, various studies were conducted to evaluate matrix effects and mostly focused on clean-up procedures, matrix types, and ion selection [30,31,32]. Two approaches were used for reducing the concentrations of interfering compounds. One approach involves the use of lower injection volumes, while the other approach involves sample dilution [33]. To the best of our knowledge, only a few studies compared matrix effects by using different injection volumes. This approach was also combined with sample dilution methods. However, there are no studies investigating numerous pesticides using a combination of small injection volume and dilution technique. The objective of our research was to develop a multi-residue analysis method for mandarin orange and grapefruit by optimizing the sample preparation procedure and injection volume, and performing sample dilution to achieve the desired recovery values and matrix effects.

## 2. Materials and Methods

### 2.1. Chemicals and Reagents

A total of 235 analytical pesticide standards in the form of stock solutions (1000 μg/mL) were purchased from AccuStandard (New Haven, CT, USA). Moreover, 5, 33, 2, 9, and 3 analytical pesticide standards in powder form were purchased from Wako Pure Chemical Industries (Osaka, Japan), Dr. Ehrenstorfer (Augsburg, Germany), ChemService (West Chester, PA, USA), Sigma-Aldrich (St. Louis, MO, USA), and Dongbang Agro Corporation (Republic of Korea), respectively. Acetonitrile (HPLC grade), methanol (HPLC grade), acetone (HPLC grade), and formic acid (LC–MS grade) were purchased from Merck (Darmstadt, Germany). Ammonium formate (LC–MS grade) was purchased from Sigma-Aldrich. Purified water was prepared in the laboratory using the Automatic purification system (Autwomatic Plus GR; Wasserlab, Navarra, Spain). The QuEChERS AOAC 2007.01 method extraction kit (1.5 g sodium acetate and 4 g magnesium sulfate), EN 15,662 method extraction kit (1 g sodium chloride, 4 g magnesium sulfate, 1 g sodium citrate, and 0.5 g disodium citrate sesquihydrate), type A dispersive solid-phase extraction (d-SPE) kit (150 mg MgSO_4_ and 25 mg primary secondary amine (PSA)), type B d-SPE kit (150 mg MgSO_4_, 25 mg PSA, and 25 mg octadecylsilane (C18)), type C d-SPE kit (150 mg MgSO_4_, 25 mg PSA, and 7.5 mg graphitized carbon black (GCB)), and type D d-SPE kit (150 mg MgSO_4_ and 50 mg PSA) were purchased from Agilent Technologies (Santa Clara, CA, USA). The blank mandarin orange and grapefruit samples were purchased from an environmentally friendly agricultural product market, Chorocmaeul Co., Ltd. (Jeonju, Korea).

### 2.2. Stock Solution Mixtures and Matrix-Matched Standard Solutions

The stock solutions of individual powder form standards with a concentration of 1000 μg/mL were prepared in acetonitrile, acetone, or methanol. The stock solution mixtures (2 μg/mL) were prepared by combining individual stock solutions. The solutions with concentrations of 2.5–250 μg/L were prepared by serial dilution of the stock solution mixtures with acetonitrile to obtain a standard calibration curve. To construct a matrix-matched calibration curve, the solutions were prepared using blank mandarin orange and grapefruit sample extracts using the same procedures as those used for the recovery test. All stock and working solution mixtures were stored at −20 °C until use. 

### 2.3. Sample Preparation and Instrument Condition Optimization

To achieve optimum recovery and matrix effects, the extraction solvents (acetonitrile and 0.1% formic acid in acetonitrile), partitioning salts (QuEChERS extract kits AOAC and EN), and d-SPE kits (type A, type B, type C, and type D) were optimized through recovery tests at a concentration of 50 μg/mL. After optimization of sample preparation, four different injection volumes (1, 2, 5, and 10 μL) were compared to confirm the peak shape and matrix effect. 

### 2.4. Instrument Condition

The LC–MS/MS analysis of 287 compounds was conducted using an AB SCIEX Triple Quad^TM^ 5500 coupled with an Exion LC^TM^ (SCIEX, Redwood City, CA, USA). A Halo C18 column (2.1 × 150 mm, particle size: 2.7 μm) was used for separating analytes at an oven temperature of 40 °C with two mobile phases consisting of (A) 0.1% formic acid and 5 mM ammonium formate in water and (B) 0.1% formic acid and 5 mM ammonium formate in methanol. A 20 min gradient was performed at a flow rate of 0.2 mL/min with 5% of the initial mobile phase B, which was held for 0.2 min, ramped to 50% for 0.3 min, increased linearly to 90% for 9 min, raised to 98% for 4 min, and then held at 98% for 3.5 min. To reach a mobile phase equilibrium condition, B was decreased to 5% for 0.1 min and held for 2.9 min. Under MS/MS conditions, the ESI positive and negative modes were employed for sample analysis. The pressures of the curtain gas, collision gas, nebulizer gas, and drying gas were 25, 10, 50, and 50 psi, respectively. The temperature of the ion source was 550 °C, and the positive and negative ion spray (IS) voltages were +5500 V and −4500 V, respectively. To achieve optimum sensitivity and selectivity, a scheduled multiple reaction monitoring mode was employed for all pesticides, and data processing was conducted using the MultiQuant^TM^ 3.0.2 software (version number: 3.0.8664.0, SCIEX). 

### 2.5. Sample Preparation 

The peeled blank mandarin orange or grapefruit samples were homogenized and 10 g of the sample was weighed and placed in a 50 mL centrifuge tube. The samples were shaken vigorously using a Geno/Grinder homogenizer (SPEX SamplePrep, Metuchen, NJ, USA) for 1 min before adding 0.1% formic acid in acetonitrile. Then, the QuEChERS AOAC 2007.01 method kit (1.5 g NaOAC, 6 g MgSO_4_) was added and shaken vigorously for 1 min, and the samples were centrifuged at 3500 rpm for 5 min using a Combi-514R centrifuge (Hanil Science Co., Ltd., Incheon, Korea). Subsequently, 1 mL of the supernatant was transferred into the d-SPE tube (150 mg MgSO_4_ and 25 mg PSA) and mixed for 1 min before centrifugation at 12,000 rpm for 5 min. Finally, the supernatants were matrix-matched with acetonitrile (4:1) for LC–MS/MS analysis.

### 2.6. Matrix Effect Comparison by Dilution Approach 

To evaluate the matrix effect after reducing the matrix load, calibration solutions with different matrix fractions (5–50 μg/L) were prepared using dilution factors of 0.8, 0.5, 0.2, 0.1, and 0.05, which were calculated as follows:Dilution factor = *V*_B_/*V*_S_ + *V*_A_ + *V*_B_(1) where *V*_B_ is the volume of the blank sample extract, *V*s is the volume of the solvent standard, and *V*_A_ is the volume of acetonitrile. *V*s is 0.05 mL, and the sum of *V*s + *V*_A_ + *V*_B_ is 1 mL. Consequently, different blank sample extracts were diluted with acetonitrile.

### 2.7. Method Validation

The analytical method was validated for parameters including selectivity, the limit of quantitation (LOQ), linearity, accuracy, precision, and matrix effect. A schematic diagram of the entire experiment is shown in Figure 1. To determine the selectivity of each target analyte, the blank mandarin orange and grapefruit sample extracts were tested. LOQ was set as the lowest validated level with sufficient recovery and precision. The linearity (*n* = 6) of the matrix-matched calibration curve was evaluated as the coefficient of determination (*R*^2^). The accuracy and precision of the recovery were evaluated by the SANTE/2020/12830 guideline criteria (70–120%, relative standard deviation (RSD) ≤ ±20%) at fortification levels of 0.001, 0.0025, 0.005, 0.01, and 0.05 mg/kg (*n* = 3). The matrix effect of each compound was calculated as follows [23]:Matrix effect (%) = (slope of the calibration curve obtained using matrix-matched solution/slope of the calibration curve obtained using pure solvent − 1) × 100(2)

## 3. Results and Discussion

### 3.1. Optimization of Sample Preparation

In a previous study, several preparation parameters were evaluated to achieve the highest recoveries and proper matrix effect for the simultaneous analysis of pesticides [34]. These parameters include different solvents [35,36], different partitioning salts [37], and different d-SPE sorbents [38]. To evaluate the extraction efficiency, 0.1% formic acid was added to acetonitrile to avoid the degradation of certain analytes [35]. Furthermore, to evaluate the effects of different partitioning salts, the AOAC 2007.01 and buffered EN 15662 standard methods utilized by international regulatory agencies were applied [37,39]. These QuEChERS standard methods are used for the analysis of pesticide residues in various food matrices such as spinach [40], strawberry, passion fruit, pineapple, grapes [41], plum, pear, cherry, nectarine, apricot, apple, and quince [42]. After partitioning, the d-SPE cleanup procedure was optimized to eliminate matrix effects and avoid contamination due to LC hardware [43]. The main matrices in mandarin orange and grapefruit are water and sugar. Therefore, MgSO_4_ and PSA sorbent-based d-SPE tubes were compared, such as type A d-SPE (MgSO_4_ 150 mg and PSA 25 mg), type B d-SPE (MgSO_4_ 150 mg, PSA 25 mg, and C18 25 mg), type C d-SPE (MgSO_4_ 150 mg, PSA 25 mg, and GCB 7.5 mg), and type D d-SPE (MgSO_4_ 150 mg and PSA 50 mg). MgSO_4_ and PSA adsorbs to water and sugar, C18 adsorbs fatty acids, and GCB adsorbs pigments [38,44]). No significant differences are observed in the number of pesticides; however, more pesticides are recovered using the 0.1% formic acid solution in acetonitrile, AOAC method, and type A d-SPE tubes, which satisfy the recovery criteria established for mandarin orange and grapefruit. We also find that the average recoveries are closer to 100% when type A d-SPE is used. For instance, the lower recovery of thiabendazole in observed when using type C d-SPE, which is a well-known planar pesticide. In addition, the recovery of thidiazuron also decreases when the GCB-containing d-SPE tube is utilized, because GCB easily absorbs aromatic compounds including phenylurea [38]. When using type D d-SPE, the sulfonylurea pesticides, such as azimsulfuron, chlorimuron-ethyl, and flucetosulfuron, have lower recoveries due to adsorption on PSA. These compounds contain carboxyl (-COOH) or acidic sulfonamide (-SO_2_NH-) groups, which may interact with amine groups on PSA sorbent [45]. In a previous study, a similar phenomenon was observed for bensulfuron-methyl, cyclosulfamuron, ethametsulfuron-methyl, ethoxysulfuron, thifensulfuron-methyl, and tribenuron-methyl [38]. Some improvements compared to previous studies are achieved in our study with respect to sample preparation (Table 1).

### 3.2. Injection Volume Optimization

To gain the proper matrix effect, best peak shape, and sensitivity during the LC–MS/MS analysis, the injection volume is one of the most important parameters [33,46,47]. In the previous studies, the injection volumes were optimized for the above-mentioned reasons. An injection volume of 5 μL was selected in the case of spinach analysis using Shimadzu LC–MS/MS because a high injection volume such as 10 μL could lead to the suppression of the linearity of the calibration curve [33]. The sensitivity is not linearly proportional to injection volume, and too large an injection volume may lead to distorted peaks, carry-over, and broad peak widths [46,47]. In the case of cabbage and rice analyses using Thermo Scientific LC–MS/MS, an injection volume of 6 μL was selected to achieve optimum peak shape [47]. An injection volume of 20 μL was selected to provide the highest sensitivity in the case of milk and milk product analyses using Waters LC–MS/MS [48]. Therefore, it is necessary to optimize the injection volume. 

In this study, the sensitivity, peak shape, and matrix effect were evaluated at injection volumes of 1, 2, 5, and 10 μL. Results show that an injection volume of 2 μL ensures sensitivity, good peak shape, and proper matrix effect. For injection volumes of 5 and 10 μL, the peak shapes are distorted for both mandarin orange and grapefruit (Figure 2). When comparing injection volumes of 5 and 2 μL, the number of pesticides in mandarin orange that are beyond the matrix effect range (−20–20%) increases significantly (Figure 3). When comparing injection volumes of 1 and 2 μL, the latter results in greater sensitivity. Hence, an injection volume of 2 μL was selected for further analysis. We evaluated the matrix effect in more detail to that in previous studies (Table 1). For mandarin orange, Morris and Shriner used a different injection volume (5 μL) from our study probably because of different instruments, column types, compounds, or mobile phases [49]. When using two columns, which are of the same length and contain particles of the same size but have different inner diameter, a larger injection volume is needed to obtain the same sensitivity. Thus, it is important to examine if the selected injection volume can afford sufficient sensitivity for quantification. Besil et al. also analyzed pesticides in mandarin orange [50]. For imidacloprid, they obtain a matrix effect that is outside the range of −20–20% using an injection volume of 5 μL. This is consistent with our study, in which the matrix effect is −25.0% using the same injection volume. On the other hand, we obtain a matrix effect of 2.53% using an injection volume of 2 μL. Hence, it is worth optimizing the injection volumes to develop a more reliable method.

### 3.3. Matrix Effect Evaluation before and after Dilution

Matrix effects change analyte responses and, ultimately, the overall analysis results [23]. Sample dilution is a simple and effective approach to inject fewer matrix components into the instrument, thereby diminishing matrix effects and extending the life of the instruments [21]. The SANTE/2020/12830 guidelines allow the use of calibration curves with standards in solvent only if validation experiments demonstrate that the matrix effects are insignificant (≤±20%). To confirm whether they can be quantified with a solvent standard, matrix effects were evaluated by comparing the slopes of matrix-matched calibration curves (dilution factor: 0.8, 0.5, 0.2, 0.1, 0.05), which contain different percentage of matrix components and calibration curve constructed using pure standards to demonstrate that matrix effects are insignificant. There is no significant change in the number of pesticides for which the matrix effect satisfies the established range (−20–20%). However, the matrix effect of each compound changes, and three cases of the matrix effect phenomenon occur in each compound (Figure 4). In the case of fenoxycarb, when the ionization of the analyte is not affected by the matrix effect, it is found that the concentration of the analyte in all diluted samples is approximately the same; variations occur only due to random errors. In the case of bromacil, the matrix effect is eliminated by sufficient sample dilution [20]. For fluopyram, the matrix effect is outside the established range (−20–20%) by one or all dilutions. Therefore, the sample dilution approach does not achieve a negligible matrix effect for each compound. Eventually, we selected a 4:1 matrix-matched calibration corresponding to a dilution factor of 0.8. Compared to previous studies, our study shows some improvements by presenting more cases of changes in the matrix effect (Table 1). Kecojević et al. conducted a study of 179 pesticides in cabbage and rice using 1:1 matrix-matched calibration (i.e., dilution factor of 1), which indicates that dilution is sufficiently effective in reducing the matrix effect [46]. However, our results show that 4:1 matrix-matched calibration reduces the matrix effect better than 1:1 matrix-matched calibration. This indicates that dilution does not reduce the matrix effect for all pesticides, and different compounds or compounds in different samples require different dilution factors to compensate for the matrix effect. 

### 3.4. Method Validation 

The selectivity of the analytical method was determined by analyzing blank mandarin orange and grapefruit extracts. No significant interferences are detected at the same retention times of the analytes. Importantly, 100% of the compound can be quantified at LOQs lower than the PLS (0.01 mg/kg) that ranges between 0.001 and 0.01 mg/kg. To compensate for the matrix effect, the matrix-matched calibration curves (4:1) were established over the 0.5–50, 1–50, or 2.5–50 μg/L ranges, with the coefficient of determination >0.990. Recovery studies were performed for three repetitions at 1, 2.5, 5, and 10 μg/L spiking levels (LOQ levels) and 50 μg/L spiking level (high). LOQ was set as the lowest validated level with sufficient recovery and precision. The recovery values for all the analytes are satisfactory, with a recovery range of 70–120% and RSD ≤ 20% for mandarin orange and grapefruit samples (Appendix A). The recovery of analytes in mandarin orange ranges from 70.4% to 119.9% at the low spiking levels, and from 70.4% to 119.5% at the high spiking level. The recovery of analytes in grapefruit ranges from 70.3% to 119.8% at the low spiking levels, and from 74.6% to 119.0% at the high spiking level. The representative chromatogram of mefenacet in mandarin orange is shown in Figure 5. The figure shows the combined spectra of 287 pesticides, chromatogram of the lowest matrix-matched standard solution, chromatogram of the lowest sample concentration (LOQ), chromatogram of the blank sample, calibration curve, and mass spectrum of mefenacet in mandarin orange. The matrix effect evaluation was conducted by comparing the slope of the matrix-matched calibration and solvent standard calibration curves at the same concentration. For more than 94.8% of the pesticides in mandarin orange and 85.4% of pesticides in grapefruit, ignorable matrix effects (within ±20%) are apparent (Figure 2). Also, 62.4% and 70.0% of the pesticides in mandarin orange and grapefruit show ion suppression, respectively. The results of the validation method are given in the Appendix A (Appendix A). There are no significant differences obtained by the types of matrices. These results indicate that our method has the appropriate accuracy for the quantification of multiple residues in mandarin orange and grapefruit.

## 4. Conclusions

Recently, compensation for matrix effects has become a key topic in the analysis of pesticide residues by mass spectrometry. Therefore, various studies have been conducted to address this issue. In this study, good peak shape and proper matrix effects are obtained by optimization of injection volumes. However, this study confirms that dilution does not reduce the matrix effect for all pesticides, suggesting that this approach may not be a powerful and broadly applicable method. Based on these results, we developed a multi-residue method for 287 pesticides in citrus fruits. Ultimately, 2 μL of injection volume and 4:1 matrix-matched calibration were used to achieve a proper matrix effect. The analytical method is validated in terms of selectivity, the limit of quantitation (LOQ), linearity, accuracy, precision, and matrix effect. The LOQ of all pesticides achieve below the PLS ranged between 0.001 and 0.01 mg/kg. The linearity of the matrix-matched calibration curve for each compound ranges from 0.0005 to 0.05 μg/mL, and its coefficient of determination (*R*^2^) is >0.990. The recovery values of all the analytes are satisfactory, with a recovery range of 70–120% and RSD ≤20% for mandarin orange and grapefruit samples. For more than 94.8% of the pesticides in mandarin orange and 85.4% of the pesticides in grapefruit, a negligible matrix effect (−20–20%) is observed. It is expected that this method will be easily implemented in the analysis of pesticide residues in citrus fruits for routine testing in laboratories. In addition, optimization of the sample injection volume and the dilution factor is expected to be utilized as an easier alternative to the use of an analyte protectant to compensate for matrix effects.

## Figures and Tables

**Figure 1 foods-11-03522-f001:**
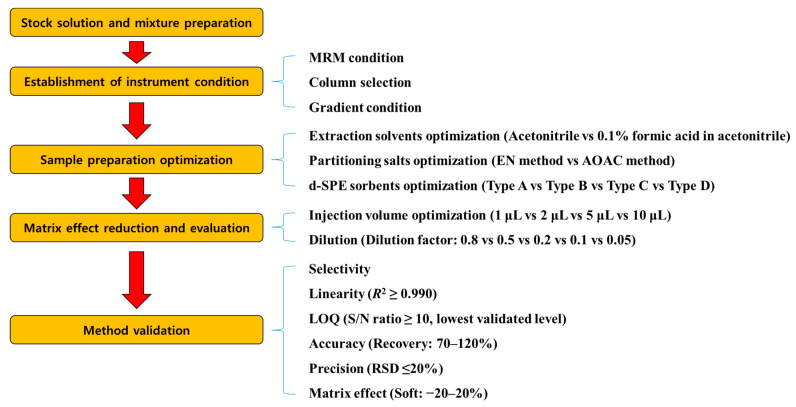
Schematics of the entire experiment.

**Figure 2 foods-11-03522-f002:**
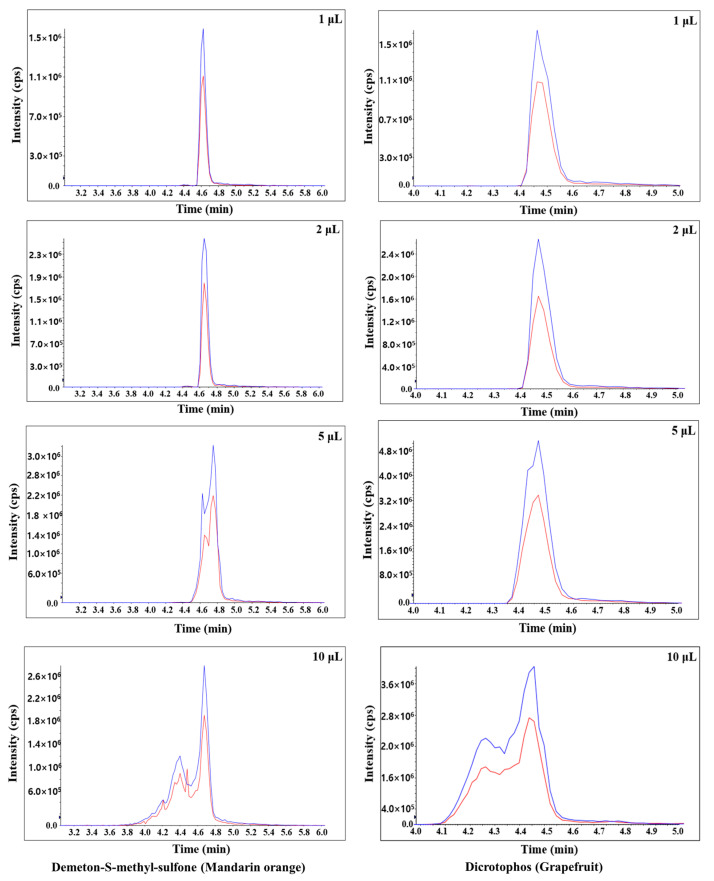
Typical chromatograms of demeton-S-methyl-sulfone and dicrotophos in mandarin orange and grapefruit, respectively, obtained using different injection volumes (1, 2, 5, and 10 μL). The blue and red lines were quantitative and qualitative ions, respectively.

**Figure 3 foods-11-03522-f003:**
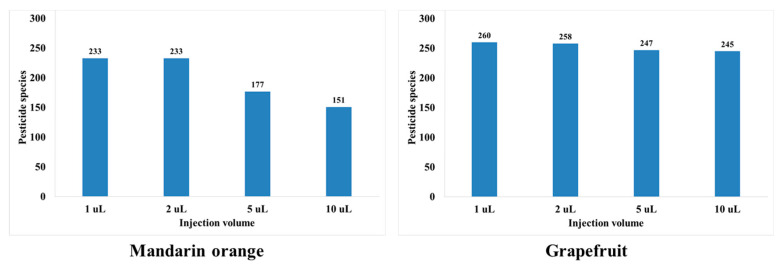
Number of pesticides showing soft matrix effect ranging from −20 to 20% depending on injection volume.

**Figure 4 foods-11-03522-f004:**
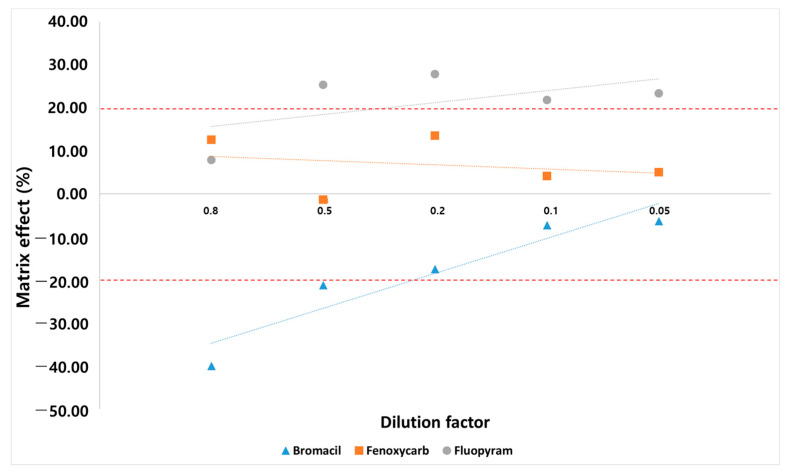
Three cases of matrix effect change affected by dilution.

**Figure 5 foods-11-03522-f005:**
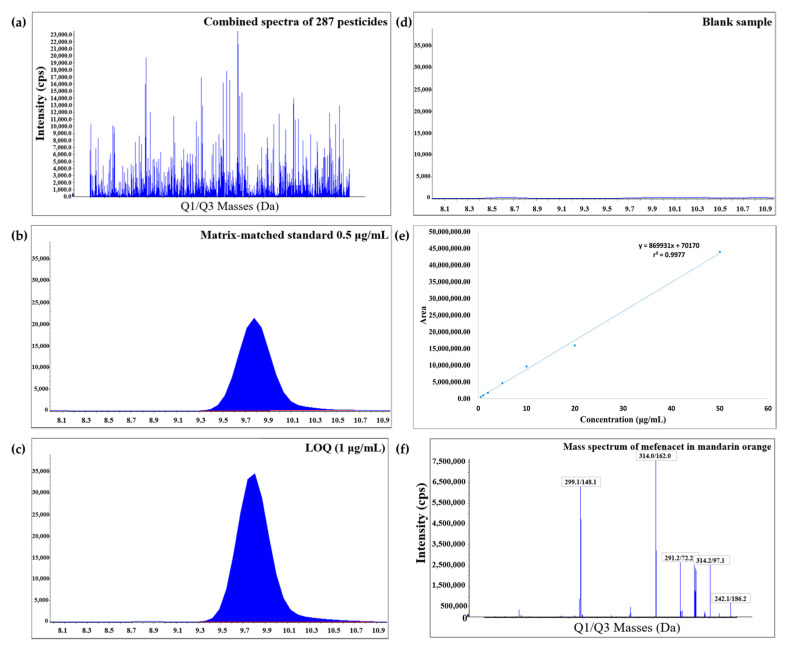
(**a**) Combined spectra of 287 pesticides. (**b**) Chromatogram of the lowest matrix-matched standard solution. (**c**) Chromatogram of the limit of quantification (LOQ). (**d**) Chromatogram of the blank sample. (**e**) Matrix-matched calibration curve. (**f**) Mass spectrum of mefenacet in mandarin orange.

**Table 1 foods-11-03522-t001:** Improvements achieved in our study compared to previous studies.

Study	Matrix	Contents	Improvements
Kang et al. [10]	Mandarin orange	100 mL extraction solvents	Less time consuming sample preparation processRequires less solvents for sample preparation
		SPE-Floril purification	
Kecojević et al. [46]	Cabbage	Compare injection volume to obtain best peak shape	Evaluation of matrix effect by injection volume
	Rice		
Ferrer et al. [27]	Orange	Reduce matrix effect by dilution for 53 pesticides	Greater number of pesticides
	Tomato		Suggests three cases of change in matrix effect
	Leak		

## Data Availability

Data is contained within the article.

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
