# Peer review of "Validation of a Multi-Residue Analysis Method for 287 Pesticides in Citrus Fruits Mandarin Orange and Grapefruit Using Liquid Chromatography–Tandem Mass Spectrometry"

_foods, 2022, doi:10.3390/foods11213522_

Round 1
Reviewer 1 Report (Previous Reviewer 2)
The revised manuscript entitled “Validation of a Multi-Residue Analysis Method for 287 Pesticides in Citrus Fruits Mandarin Orange and Grapefruit Using Liquid Chromatography-Tandem Mass Spectrometry” (Manuscript Number: foods-2004786) is rather satisfactory :
Comment:
- The comparison table should be presented to show the novelty of the work.
Author Response
Responses to the comments are attached herewith.

Reviewer 2 Report (Previous Reviewer 3)
The authors have satisfactorily addressed all the comment raised by reviewers.
Author Response
Responses to the comments are attached herewith.

Reviewer 3 Report (New Reviewer)
This paper described “Validation of a multi-residue analysis method for 287 pesticides in citrus fruit mandarin and grapefruit using liquid chromatography-tandem mass spectrometry”
The concept of manuscript is straight forward and results were clearly presented. In addition, this is a relevant and good experimental work regarding developed a LC/MS-MS method.
The work needs to be revised, especially for its innovation, introduction to partial surface analysis studies and the English of the article.
I have some observations and in my opinion a minor revision of the manuscript is required :
1. The working range in the abstract should be corrected to “0.5 – 50” μg/L. and “R2” should be written instead of “r2”.
2. Electrochemical techniques are also widely used in pesticide determinations. The advantages of the LC/MS-MS method they developed by considering the current articles below should be added to the Introduction section.
• https://doi.org/10.1016/j.jelechem.2021.115389
3. Expressions such as 0.0005, 0.001, 0.002, and 0.005 μg/mL should be written in a simpler form.
4. According to what 287 pesticides were selected. Are these pesticides used together in food sample?
5. Information on recovery or selectivity studied can be added in the article.
Author Response
Responses to the comments are attached herewith.

This manuscript is a resubmission of an earlier submission. The following is a list of the peer review reports and author responses from that submission.
Round 1
Reviewer 1 Report
1) In 2. Materials and Methods
Need Following Clarifications:
a) Page 2 of 10, Sec 2.1, Line no 95, written Acetonitrile (HPLC grade)?
Is it HPLC grade of LC-MS/MS grade? In case HPLC then need clarification vis-a-vis use of other solvents as LC-MS grade.
2) Page 3 of 10, Sec 2.2. Line no 107,
Grade of Acetone not mentioned before.
Please do as these become critical especially your manuscript is reporting for a validated analytical method to implement analysis of pesticide residues in citrus fruits for routine testing in laboratories.
2) In 3. Results and Discussion
a) Need to support your claims. Show LC-MS/MS Chromatograms/Spectras and calibration curves. Recommend to include Combined Spectra of 287 Pesticides as Figure in the main Manuscript.
b) Proof for matrix-matched calibration curves (4:1) Page no 7/10 Line no 248 and 257 Established through the ranges for 0.5, 1 or 2.5-50ul/L with the corelation coefficients from 0.990?
c) Show Ion Suppression for 62.4% and 70.0% pesticides Page no 7/10 Line no 269.
Author Response
Responses to the comments are attached herewith.

Reviewer 2 Report
In this manuscript entitled “Validation of a multi-residue analysis method for 287 pesticides in citrus fruit mandarin and grapefruit using liquid chromatography-tandem mass spectrometry” (Manuscript Number: Foods (ISSN 2304-8158)) I think it’s better to discuss about below questions. Therefore, I suggest a major revision for the manuscript.
Comments:
- Experimental: should be extended. Also, the total materials and companies, characterization of all instruments, methodology and synthesis steps should be presented with more details.
- The authors should be explaining about the importance and novelty of the work with more details.
- The text can be improved by providing a more critical discussion of related literature. For example: International journal of biological macromolecules 128, 718-723 (2019) / Arabian Journal of Chemistry 10, S3156-S3166 (2017) / International Journal of Biological Macromolecules 160, 456-469 (2020) / Polyhedron 177, 114302 (2020) / Journal of Molecular Liquids 215, 31-38 (2016)
Reviewer 3 Report
Yuan et al. have validated a multi-residue analysis method for 287 pesticides in citrus fruit mandarin orange and grapefruit using liquid chromatography-tandem mass spectrometry. It is a systematic method development study carried out with all necessary sample preparation, separation and validation parameters.
1. The usage of only “mandarin” can be misleading for a language, instead “mandarin orange” can be used throughout the manuscript.
2. R2 is not correlation coefficient, but coefficient of determination. This correction should be made throughout the manuscript.
3. Line 26-27 & 283-284, “The linearity (correlation coefficient (r2)…that ranged from 0.0005-0.05 mg/mL was higher than 0.990” should be corrected as “The linearity and coefficient of determination (r2)…ranged from 0.0005-0.05 mg/mL and >0.990, respectively”.
4. Why the word “pesticides” missing in the keywords?
5. Line 82-84, rewrite the statement for clarity.
6. Line 92, mention how many pesticide standards.
7. A schematic diagram summarizing the sequential steps of method development like optimization of sample preparation including all the methods tested, injection volume, dilution volume, matrix effect, chromatographic method development for separation and method validation.
8. Figure 1, the axis labels are not clear and should be enhanced.
9. Figure 2, the axis lines are missing and should be drawn.
10. Figure 3, the axis labels and legend labels are too small and should be enlarged for readability.
11. At the end of results and discussion, a take-home diagrammatic information on the optimized conditions evolved out of this study for this method development of multi-residue analysis of pesticides (including all that is presented in the schematic diagram according to the comment no. 7). In this way, the readers can have a quick glance of important findings.
Reviewer 4 Report
Authors prepared a well introduced and presented manuscript regarding the validation of an analytical method for the multi-residue analysis of 287 pesticides in mandarin and grapefruit. The novelty of this study is not its strong part, since numerous similar studies can be found in literature. I believe authors could revise their manuscript and address some the points that I mention below:
*Authors could rephrase the lines 53-55: “Until now, several … Panax ginseng [15], soil [16].” and refer to biological samples instead of human urine and human serum.
* Purified water was obtained in the laboratory or produced every day in the laboratory by the Automatic purification system? Please provide model of this system.
* References are double numbered.
* Authors could provide a typical chromatograph of a standard solution (in the lowest concentration), a spiked solution (in the lowest concentration) and a sample (positive). This info could mentioned in text and provided as supplementary material, if not in the main manuscript.
* Why authors choose mandarin and grapefruit as targeted samples? In other fruit or vegetables these info are not useful? Please discuss according literature.
* Is there any possibility in one sample to determine 287 different compounds? I mean a specific cultivation may be affected by specific chemicals? I believe 287 are a huge amount of pesticides that some of them will never be detected in mandarin/grapefruit samples. Is that right? Authors could discuss this aspect according literature and provide arguments regarding the needing of simultaneous determination of 287 pesticides in one fruit.
Round 2
Reviewer 4 Report
Although authors revised the manuscript according some of the recommendations, some important issues are still unanswered/ not included in revised manuscript:
* where is the novelty of this study? reply and also mention in text (important)
* discuss results with relevant literature is missing (should be added in order to support their results) - advised before in point 6
* the possibility to determine 287 different compounds in one sample is not included in text as advised before in point 6
Author Response
Responses to the comments of Reviewer 4
Point 1: where is the novelty of this study? Reply and also mention in text (important)
Response 1: We thank you for this question. Unlike previous studies, we extensively investigated the roles of injection volume and dilution factor in compensating for matrix effects for a large number of pesticides. Our findings demonstrated the importance of optimizing these two factors, which has been neglected in previous work. (Lines: 88–89, 92–95, 317–324, 335–337 we have discussed these points in the revised manuscript).
Point 2: discuss results with relevant literature is missing (should be added in order to support their results) –advised before in point 6
Response 2: We thank you for this suggestion. (Lines: 240–248, 275–283 we have compared our results with the literature to support our conclusions).
Point 3: the possibility to determine 287 different compounds in one sample is not included in text as advised before in point 6
Response 3: We thank you for noting this. (Lines: 36–37, 48–52 we have made the necessary revision).
*Thank you for your comments. They provided an opportunity to further improve our manuscript.